# Multi-Hop Clustering and Routing Protocol Based on Enhanced Snake Optimizer and Golden Jackal Optimization in WSNs

**DOI:** 10.3390/s24041348

**Published:** 2024-02-19

**Authors:** Zhen Wang, Jin Duan, Pengzhan Xing

**Affiliations:** 1School of Electronic Information Engineering, Changchun University of Science and Technology, Changchun 130022, China; wangzhen1531002@163.com; 2Company’s R&D Department, Changchun Guanghua Micro-Electronics Equipment Engineering Center Co., Ltd., Changchun 130022, China; xpz35@126.com

**Keywords:** wireless sensor networks, snake optimizer, golden jackal optimization, energy efficient, multi-hop, routing

## Abstract

A collection of smaller, less expensive sensor nodes called wireless sensor networks (WSNs) use their sensing range to gather environmental data. Data are sent in a multi-hop manner from the sensing node to the base station (BS). The bulk of these sensor nodes run on batteries, which makes replacement and maintenance somewhat difficult. Preserving the network’s energy efficiency is essential to its longevity. In this study, we propose an energy-efficient multi-hop routing protocol called ESO-GJO, which combines the enhanced Snake Optimizer (SO) and Golden Jackal Optimization (GJO). The ESO-GJO method first applies the traditional SO algorithm and then integrates the Brownian motion function in the exploitation stage. The process then integrates multiple parameters, including the energy consumption of the cluster head (CH), node degree of CH, and distance between node and BS to create a fitness function that is used to choose a group of appropriate CHs. Lastly, a multi-hop routing path between CH and BS is created using the GJO optimization technique. According to simulation results, the suggested scheme outperforms LSA, LEACH-IACA, and LEACH-ANT in terms of lowering network energy consumption and extending network lifetime.

## 1. Introduction

A WSN is a network made up of numerous sensor nodes dispersed within a certain region. To enable data collection, transfer, and analysis, these nodes can be wirelessly connected to one another [1,2]. WSNs are becoming more and more appealing for a wide range of application fields, including military applications, emergency response and security, environmental monitoring, health and medical, automated industrial control, smart cities, transportation and logistics, and entertainment [3,4,5]. WSNs’ low cost and ease of deployment have led to their widespread application in many different industries. But energy is scarce and sensor node batteries are hard to replace. Therefore, it is imperative to address the issue of the energy usage of networks and increase the network’s lifespan [6,7].

Given that the primary cause of node energy consumption is wireless communication [8], energy-efficient communication protocols are essential for drastically reducing network energy consumption and increasing network lifetime. Of these protocols, the clustering routing protocol is a notable example of an essential subset in the field of energy-efficient WSN communication protocols. It has proven to be a very successful method for energy conservation in WSNs with limited energy supplies. Organizing sensor nodes into multiple clusters, each with a CH and additional common nodes, is the core idea behind the cluster-based routing protocol. The common nodes in each cluster gather information and send it to the CH. Following data collection, the regular nodes’ data packets are received, combined, and processed by the CHs, which subsequently forward the processed and combined data to BS. The clustered routing protocol is essential for WSNs’ energy conservation since it reduces energy usage and helps manage overhead traffic [9,10,11,12].

The key idea of network clustering is intimately related to how well the clustered routing protocol affects network energy consumption. As communication distance reduces, so does the amount of energy used for communication inside clusters. Given that CHs use more energy to perform functions like receiving, fusing, and transferring data than normal nodes, choosing the right CHs is essential for enhancing network load balancing. Furthermore, the utilization of a multi-hop communication technique is efficacious in mitigating communication energy consumption, particularly in light of the high energy expenditure linked to long-distance communication. This work employs the SO optimization technique to solve the NP-hard problem [13,14,15,16,17] of optimal CH selection, and the best route between CHs is determined during the data transmission phase using the GJO optimization technique. Below is a summary of this research work’s main contributions:(1)The ESO-GJO method adds a Brownian motion function to the SO algorithm’s exploitation stage in order to avoid local optima.(2)The proposed approach uses SO to select the optimal CHs and considers variables such as energy and distance when building the fitness function.(3)The suggested algorithm uses GJO to determine the route from CH to BS and initializes the GJO population through backward learning.

The structure of the rest of the article is summarized below:Section 2: Summarizes previous work in related fields.Section 3: Introduces the energy model and network model.Section 4: Offers a detailed introduction to the algorithm proposed in this article.Section 5: Performance metrics simulation demonstration and comparison.Section 6: Summarizes the research work in this paper and suggests future work to be carried out.

## 2. Related Works

Numerous routing methods have been developed with the goal of increasing energy efficiency and lengthening network lifetime in order to address the problem of limited energy in WSNs. LEACH [18] is one of the most well-known clustering routing methods for WSNs. Each iteration of the LEACH protocol is divided into two stages: clustering and stabilization. During the clustering phase, each node generates a random integer between 0 and 1, which it subsequently compares to the value obtained from a threshold calculation. The node is not elected as CH in the following round if the random number is less than the threshold and it was not elected in the preceding 1/p (*p* is the proportion of nodes that became CH in the current round) rounds. During the stabilization phase, the sensor nodes transmit the information they have collected to the CH. After completing data fusion on the gathered data, CH sends the information in a single hop to the aggregate node. The CH needs more energy to run because it has a higher mission load than other nodes, which is a major drawback of the LEACH algorithm. The energy-efficient multi-hop routing protocol L-LEACH (Energy Balanced) [19] solely takes the distance component into account. Like LEACH, L-LEACH is divided into two phases. Every round, the energy load is split equally among the network nodes and new CHs are chosen. Nonetheless, a noteworthy constraint of the algorithm is its disregard for the algorithm’s intricacy and the corresponding network overhead. This oversight could be considered a negative in assessing the overall performance of L-LEACH. In order to send data from orphan nodes in the network that cannot be fully covered to the BS, Jerbi et al. [20] designed the O-LEACH (orphan-LEACH) protocol. The protocol proposes two solutions. One method is to have a cluster member serve as a gateway, enabling non-clustered nodes to join the cluster by designating that gateway node as the CH, so that the data of the isolated nodes can be sent through the gateway node aggregation to the CH of the cluster where the gateway node is located. Another solution is that cluster members can form subclusters with orphan nodes, and the first orphan node to join will serve as the cluster head for that subcluster. It is important to remember that this approach works only in situations when there is a finite number of cluster members. The Multi-Hop LEACH (MH-LEACH) protocol was created by Neto et al. [21] to find the best pathways to the BS via several intermediate CHs. According to this protocol, each CH broadcasts an advertisement message in order to build its routing table according to the strength of the signal it receives. However, they are forwarded to the CH to be checked for possible looping or an opposite direction before employing these routes. C Long et al. [22] and Tanushree Agarwal et al. [23] propose LEACH-ANT and LEACH-IACA protocols, respectively, both of which are based on LEACH for cluster head selection and utilize an ant colony algorithm to find the optimal path in the inter-cluster transfer phase. However, the algorithm suffers from local optimization and slow convergence speeds.

Swarm intelligence has a lot of potential for WSN routing optimization. In order to extend the network life cycle, Gobi Natesan et al. [24] proposed a clustering algorithm combining Mayfly and Aquila called MFA-AOA. The algorithm uses MFA for CH selection using variables like energy and distance, among others, to build the fitness function and choose CHs. The algorithm uses AOA to select the optimal route between CH and BS. But the algorithm does not adequately consider the complexity. Y. Yao et al. [25] presented the IAOAR routing protocol, which is based on the Archimedean optimization technique and can reduce energy efficiency. The algorithm selects the optimal CH based on AOA. Based on variables such the CH’s residual energy, the distance between clusters, and the CH’s distance from the BS, IAOAR builds the fitness function for choosing the best CHs.To determine the shortest path between CHs, the algorithm employs an ant colony optimization technique. However, it is important to remember that this approach does not handle network heterogeneity, which could be considered in future enhancements. An energy-efficient clustering technique, called EPOA-CHS (Enhanced Pelican Optimization Algorithm for Selecting Optimal CH Set), was proposed by Wang, Z. et al. [26]. The algorithm uses POA for optimal CH selection. EPOA-CHS builds the fitness function for choosing the CH set using characteristics like node energy and inter-cluster distance. However, EPOA-CHS is designed as a single-hop clustering protocol, which means that it can only operate within single-hop communication. This characteristic may affect its applicability in specific network scenarios where multi-hop communication is more advantageous. Punithavathi et al. [27] introduced BWO-IACO, an energy-efficient clustering approach based on the combination of Ant Colony Optimization and Black Widow Optimization. Using BWO, the algorithm chooses a suitable set of CHs. The fitness function used to choose CHs is composed of several criteria, including node centrality, node degree, and intra-cluster. The algorithm uses improved ACO for inter-cluster routing. However, the technique does not account for how data aggregation affects data transmission energy. Vinitha et al. [28] suggested a multi-hop routing protocol with Cat Swarm Optimization (CSO) and Salp Swarm Optimization (SSA) to lower the energy consumption and latency of wireless sensor networks. The algorithm uses LEACH to select CHs and C-SSA to select the optimal hops. Energy, distance, and latency are among the elements that make up the fitness function used to choose the best hops. However, the applicability of the algorithm to large-scale networks has not been fully considered, which may be a topic for development or further research. To minimize network power consumption, Senthil, G.A. et al. [29] proposed a routing protocol capable of efficient cluster head election and path optimization. The protocol combines Lightning Search Algorithm and Particle Swarm Optimization and is called PSO-LSA. However, the protocol focuses too much on orphan nodes, which may lead to degradation of the performance of the protocol.

## 3. Preliminaries

This section furnishes details regarding the network model and energy model.

### 3.1. Network Model

Figure 1 illustrates the structure of the WSNs.

The following factors are taken into account when creating the network model:In WSNs, every sensor node has the same initial energy.Distances between nodes are calculated using Euclidean distance.Within the sensing area, sensor nodes are haphazardly placed and maintain a fixed location once deployed.The BS can be positioned either within or outside the screening region, and it remains stationary.The sensor node as a cluster member transmits the collected environmental data to the CH.The aggregated data are transferred to the BS via CH.

### 3.2. Energy Model

The standard first-order radio model, as described in [30,31], is the foundation for this article’s estimation of the nodes’ energy consumption. Equation (Equation 1) gives the transmitting node’s energy usage, which is referred to as $E_{TX}\left(l,d\right)$. This computation is useful when a node sends *l* bits of data to a receiving node that is located *d* away. Furthermore, Equation (Equation 2) establishes the receiving node’s energy consumption, which is represented by the symbol $E_{RX}\left(l\right)$.
(1)$$E_{TX}\left(l,d\right)=\left\{\begin{array}{c} lE_{elec}+l\epsilon_{fs}d^{2},d\leq d_{0}\\ lE_{elec}+l\epsilon_{mp}d^{4},d>d_{0} \end{array}\right.$$
(2)$$E_{RX}\left(l\right)=lE_{elec}$$

Here, $\epsilon_{fs}$ and $\epsilon_{mp}$, respectively, represent the energy consumption coefficients of the amplifier circuit for the free-space fading model and the multipath fading model. Furthermore, the transmission and receiving circuits’ energy consumption coefficient is denoted by $E_{elec}$. Equation (Equation 3) establishes the threshold, which is represented as $d_{0}$ and acts as the border between the two attenuation models. The transmission uses the free-space fading model if the distance d is less than $d_{0}$, which results in an energy consumption of ${l\epsilon }_{fs}d^{2}$ for the amplifier circuit. On the other hand, the multipath fading model is applied where *d* > $d_{0}$, resulting in an energy consumption of ${l\epsilon }_{mp}d^{4}$ for the amplifier circuit.
(3)$$d_{0}=\sqrt{\frac{\epsilon_{fs}}{\epsilon_{mp}}}$$

Data are received by the CH from the cluster, and it uses some energy to transfer the aggregated data to the BS. Consequently, it becomes imperative for the CH to possess a sufficient reservoir of energy to effectively relay data to the BS. The energy level of the CH is represented by the letter $E_{CH}$,
(4)$$E_{CH}=l\cdot \frac{n}{k}\cdot \left(E_{elec}+E_{DA}\right)+l\cdot \varepsilon_{mp}\cdot d_{toBS}^{4}$$
where $E_{elec}$ is the energy used by the transmitter. $E_{DA}$ is the energy necessary for data aggregating, and a data packet is represented by the symbol *l*. When transmitting over long distances to a base station, as shown by $d_{toBS}^{4}$, the transmitter amplifier’s energy usage is represented by $\epsilon_{mp}$. The energy consumption of non-cluster heads is shown by $E_{NCH}$.
(5)$$E_{NCH}=l\cdot E_{elec}+l\cdot \varepsilon_{fs}\cdot d_{toCH}^{4}$$

Within a specific context, the symbol *l* denotes a data packet, $E_{elec}$ denotes the transmitter’s energy consumption, $\epsilon_{fs}$ denotes a transmitter amplifier in a free condition, $d_{toBS}^{2}$ denotes the CH’s short distance, and $d_{toBS}^{4}$ denotes the BS’s long distance.

## 4. Proposed Method

We will provide a detailed introduction to the proposed ESO-GJO protocol in this section. Each iteration of the ESO-GJO protocol encapsulates both the CH selection phase and the subsequent data transmission stage, amalgamating these crucial processes within every round. In the cluster head selection phase, we use the SO optimization algorithm for cluster head selection and in the data transmission phase, we use the GJO optimization algorithm for optimal path selection. In addition, the fitness function plays a crucial role in the SO algorithm and GJO, which is an indicator of the degree of superiority of a solution, and provides a search direction for the algorithm. Through the rational design of the fitness function, we can help the algorithm to find the optimal solution of the problem faster. The choice of fitness function should be determined according to the needs of the specific problem, and it can be defined based on the constraints of the problem and the objective function. Figure 2 shows the flow block diagram of the ESO-GJO algorithm.

### 4.1. CH Selection Stage

#### 4.1.1. CH Selection Using SO

In this section, we describe how to use the SO [32] optimization algorithm for optimal cluster head set selection for wireless sensor networks. The suggested algorithm’s main goal is to carefully choose CHs from the pool of regular sensor nodes, strategically integrating considerations for energy conservation. This overarching aim is geared towards proactively elongating the network’s operational lifetime.

An individual in SO symbolizes a whole solution. It represents the ideal locations of the CHs for the proposed algorithm’s CH selection. Let $X=\left[X_{i,1}\left(t\right),X_{i,2}\left(t\right),X_{i,3}\left(t\right),\ldots ,X_{i,D}\left(t\right)\right]$ represent the *i*th member of the population, and let $X_{\left(i,D\right)}\left(t\right)=\left(x_{i}d\left(t\right),y_{i}d\left(t\right)\right),1\leq i\leq N,$1≤d≤D represent each component’s coordinates for the sensor nodes chosen to serve as CH. Every single person has the same number of dimensions represented by the letter *D*; this corresponds to the quantity of CH.

The snakes choose a random position during the exploration phase and adjust their position in relation to it if Q<0.25. The following steps are followed to mimic the exploration phase:(6)$$X_{i,m}\left(t+1\right)=X_{rand,m}\left(t\right)\pm c_{2}\times A_{m}\times \left(\left(X_{max}-X_{min}\right)\times rand+X_{min}\right)$$

The *i*th male position is shown by $X_{i,m}$, the random male position is indicated by $X_{rand,m}$, and the male’s ability to find food is indicated by $A_{m}$. The calculation for this is as follows:(7)$$A_{m}=exp\left(\frac{-f_{rand,m}}{f_{i,m}}\right)$$
where $c_{2}=0.05$ is defined, the fitness of $X_{rand,m}$ is denoted by $f_{rand,m}$, and the *i*th member of the male group’s fitness is denoted by $f_{i,m}$.
(8)$$X_{i,f}=X_{rand,f}\left(t+1\right)\pm c_{2}\times A_{f}\times \left(\left(X_{max}-X_{min}\right)\times rand+X_{min}\right)$$
where $A_{f}$ is the female’s capacity to locate food and may be computed as follows; a random female’s position is indicated by $X_{rand,f}$; and position of the *i*th female is indicated by $X_{i,f}$.
(9)$$A_{f}=exp\left(\left(-f_{\left(rand,f\right)}\right)/f_{\left(i,f\right)}\right)$$

In this case, the fitness of $X_{rand,f}$ is represented by $f_{\left(rand,f\right)}$, and the *i*th member of the female group’s fitness is denoted by $f_{i,f}$.

The constant coefficient $c_{3}$ is utilized to update the snake’s position throughout the exploiting phase. In this study, instead of using the constant coefficient $c_{3}$, we use the Brownian motion function to keep the algorithm from reaching a local optimum. One of many physical processes in which a quantity continuously fluctuates between tiny, random fluctuations is called Brownian motion. It is believed that Brownian motion is a Markov process coupled with a Gaussian process. The normal (Gaussian) equation was altered by the algorithm’s changes to the size and number of agents, which are used to determine the Brownian motion steps, and the Brownian motion was finalized (Equation (Equation 10))
(10)$$f\left(\delta \right)=\frac{1}{\sqrt{2\pi \sigma }}exp\left(-\frac{{\left(\delta -\mu \right)}^{2}}{2\sigma^{2}}\right)$$

The following formula can be used to define the temperature Temp.
(11)$$Temp=exp\left(\frac{-t}{T}\right)$$

Here, *t* represents the current iteration, and *T* signifies the maximum number of iterations.

The definition of the quantity of food is *Q*. The following formula can be used to determine how much food is consumed.
(12)$$Q=c_{1}*exp\left(\frac{t-T}{T}\right)$$
where $c_{1}=0.5$. If the temperature>0.6, snakes only travel in the direction of food.
(13)$$X_{\left(i,j\right)}\left(t+1\right)=X_{food}\pm f\left(\delta \right)\times Temp\times rand\times \left(X_{food}-X_{\left(i,j\right)}\left(t\right)\right)$$
where $X_{food}$ represents the best individual’s position and $X_{i,j}$ represents the individual’s position. The snake will go into either the fighting or mating mode if the temperature is less than 0.6.

In Fight Mode,
(14)$$X_{i,m}\left(t+1\right)=X_{i,m}\left(t\right)+f\left(\delta \right)\times FM\times rand\times \left({Q\times X}_{best,f}-X_{i,m}\left(t\right)\right)$$
where $X_{i,m}$ represents the male position, FM denotes the fighting prowess of the male agent, and $X_{best,f}$ represents the best individual’s position within the female group.
(15)$$X_{i,f}\left(t+1\right)=X_{i,f}\left(t\right)+f\left(\delta \right)\times FF\times rand\times \left({Q\times X}_{best,m}-X_{i,f}\left(t\right)\right)$$
where FF denotes the female agent’s combat prowess, $X_{i,f}$ indicates that she is the top member of the male group, and $X_{best,m}$ represents her position.

FM is represented by Equation (Equation 16) and FF is represented by Equation (Equation 17):(16)$$FM=exp\left(\frac{-f_{best,f}}{f_{i}}\right)$$
(17)$$FF=exp\left(\frac{-f_{best,m}}{f_{i}}\right)$$
where $f_{i}$ is the agent fitness, $f_{best,m}$ is the best agent of the male group, and $f_{best,f}$ is the best agent of the female group. In Mating Mode,
(18)$$X_{i,m}\left(t+1\right)=X_{i,m}\left(t\right)+f\left(\delta \right)\times M_{m}\times rand\times \left({Q\times X}_{i,f}\left(t\right)-X_{i,m}\left(t\right)\right)$$
(19)$$X_{i,f}\left(t+1\right)=X_{i,f}\left(t\right)+f\left(\delta \right)\times M_{f}\times rand\times \left({Q\times X}_{i,m}\left(t\right)-X_{i,f}\left(t\right)\right)$$
where $M_{m}\&M_{f}$ denotes the ability of the male and female to mate, respectively; $X_{i,f}$ and $X_{i,m}$, respectively, denote the positions of the *i*th agents within the female and male groups. These values can be calculated as follows:(20)$$M_{m}=exp\left(\frac{-f_{i,f}}{f_{i,m}}\right)$$
(21)$$M_{f}=exp\left(\frac{-f_{i,m}}{f_{i,f}}\right)$$

Replacement of worst males and females selected from hatched eggs:(22)$$X_{worst,m}=X_{min}+rand\times \left(X_{max}-X_{min}\right)$$
(23)$$X_{worst,f}=X_{min}+rand\times \left(X_{max}-X_{min}\right)$$
where $X_{worst,m}$ is the worst individual in the male group; $X_{worst,f}$ is the worst individual in the female group.

Finally, the globally optimal individual vector is used as the optimal CH set.

#### 4.1.2. Fitness Function of SO

The fitness function used by the SO algorithm to determine the ideal set of CHs is made up of the following variables and formulas.

(1)Energy consumption of CH:This is the first parameter of the fitness function. Our goal is to prioritize nodes with low energy consumption as CH. Let $E_{c}\left({CH}_{i}\right)$ represent the ${CH}_{i}$’s energy consumption, and assume that 1≤i≤K, *K* is the number of CHs.Consequently, minimizing the objective function $f_{1}$ is our aim:
(24)$$f_{1}=\frac{1}{K}\sum_{i=1}^{K}E_{c}\left({CH}_{i}\right)$$(2)Distance between node and BS:This is the second parameter of the fitness function. Our goal is to prioritize the node with the shortest distance to the common node and BS as the CH. Let $D\left(s_{j},{CH}_{i}\right)$ be the distance from intra-cluster node $s_{j}$ to cluster head node ${CH}_{i}$. Let $D\left({CH}_{i},BS\right)$$D\left({CH}_{i},BS\right)$ be the distance from ${CH}_{i}$ to BS where 1≤i≤K, *K* represents the quantity of CHs and 1≤j≤M, *M* represents the quantity of cluster members. Consequently, minimizing the objective function $f_{2}$ is our aim:
(25)$$f_{2}=\sum_{i=1}^{K}\frac{1}{M}\left(\sum_{j=1}^{M}D\left(s_{j},{CH}_{i}\right)+D\left({CH}_{i},BS\right)\right)$$(3)Node Degree of CH: This is the third parameter of the fitness function. High-degree nodes are given preference to become CH in the ESO-GJO CH selection process. Consequently, our goal is to maximize the objective function $f_{3}$ as follows
(26)$$f_{3}=N_{deg}\left({CH}_{i}\right)$$

Our suggested SO algorithm’s fitness function is to minimize the aforementioned $f_{1}$, $f_{2}$, and $f_{3}$ functions. As a result, the following is the fitness function for our algorithm:(27)$$f=\delta_{1}\times f_{1}+\delta_{2}\times f_{2}+\delta_{3}\times f_{3}$$
where the weights for the goal function are provided by the constants $\delta_{1}$, $\delta_{2}$, and $\delta_{3}$. The total of these constants ought to equal one. Minimizing the cumulative objective function is our aim.

### 4.2. Data Routing Stage

#### 4.2.1. Routing Algorithm Using GJO

The goal of the routing problem we will now solve is to reduce the maximum possible transmission distance between CHs in a routing path. The goal of the routing problem we will now solve is to reduce the maximum possible transmission distance between CHs in the routing path. This section describes how to use the GJO [33] optimization algorithm for optimal path selection for wireless sensor networks.

The starting population is generated by random initialization in the GJO methodology, yet this approach is vulnerable to the issue of an excessively dispersed or locally concentrated population.

Due to this circumstance, the population diversity of the classic GJO algorithm is inadequate, which significantly impairs the ensuing iterative optimization search.

Thus, in order to broaden the algorithm’s search space and boost population variety, we chose to initiate the population in this study using a backward learning technique.

The following is the definition of the reverse learning strategy:

Let $X_{i}^{t}=\left[x_{i,1}^{t},x_{i,2}^{t},\cdots ,x_{i,j}^{t},\cdots ,x_{i,d}^{t}\right]$ be the *i*th individual of the current population. The general inverse learning strategy produces a general inverse solution of $X_{i}^{t}$ as $O_{i}^{t}=\left[o_{i,1}^{t},o_{i,2}^{t},\cdots ,o_{i,j}^{t},\cdots ,o_{i,d}^{t}\right]$.
(28)$$o_{i,j}^{t}=Z\left(k\right)\left(A_{i}^{t}+B_{i}^{t}\right)-x_{i,j}^{t}$$
(29)$$Z\left(k\right)=\left[aZ(k-1)+c\right]modM$$
(30)$$A_{j}^{t}=min\left(x_{\left(i,j\right)}^{t}\right),B_{j}^{t}=max\left(x_{\left(i,j\right)}^{t}\right)$$

*M* is an enormous positive number, *a* is the multiplier of the interval $\left(0,M\right)$, and *c* is incremental. Z(0) is the seed of the random number. Z(k) approximately obeys a uniform distribution in the interval [0,M−1]; its maximum period is *M*, and $\left[Z(k)/M\right]$ approximately obeys a uniform distribution in the interval [0,1).

The two primary steps of the GJO algorithm are finding prey and encircling and attacking it. These processes represent the exploration and exploitation of populations, respectively.

(1)The stage of exploration or hunting for preyThis section proposes GJO’s exploration plan. Jackals can sense and follow their prey due to their innate instincts, yet sometimes the prey escapes and is hard to catch. As a result, the females follow the males and lead the pack in search of additional prey.
(31)$$Y_{1}\left(t\right)=Y_{M}\left(t\right)-E.\left|Y_{M}\left(t\right)-rl.Prey\left(t\right)\right|$$
(32)$$Y_{2}\left(t\right)=Y_{FM}\left(t\right)-E.\left|Y_{FM}\left(t\right)-rl.Prey\left(t\right)\right|$$
where *t* is the current iteration. Prey(t) represents the prey’s position vector, while $Y_{M}\left(t\right)$ and $Y_{FM}\left(t\right)$ represent the locations of the male and female jackals. The updated positions of the male and female jackals with respect to the prey are shown by $Y_{1}\left(t\right)$ and $Y_{2}\left(t\right)$.*E* indicates the prey’s evasive energy and is represented by Equation (Equation 33):
(33)$$E=E_{1}*E_{0}$$The prey’s initial energy level is represented by $E_{0}$, while its decreasing energy is represented by $E_{1}$.
(34)$$E_{0}=2*r-1$$
where *r* can be any integer between 0 and 1.
(35)$$E_{1}=c_{1}*\left(1-(t/T)\right)$$
where $c_{1}=1.5$; the rounds of iteration are denoted by *T* and rl is calculated by Equation (Equation 36):
(36)rl=0.05∗LF(y)LF is calculated by Equation (Equation 37):
(37)$$LF\left(y\right)=\left(0.01\times (\mu \times \sigma )\right)/\left(\left(\left|v^{\left((1/\beta )\right)}\right|\right)\right)$$
where μ and *v* are random numbers between 0 and 1, β=1.5
(38)$$Y\left(t+1\right)=\left(Y_{1}\left(t\right)+Y_{2}\left(t\right)\right)/2$$(2)The stage of exploitation or encircling and leaping for preyThe mathematical model of joint roundup of male and female jackals is shown in Equations (39) and (40):
(39)$$Y_{1}\left(t\right)=Y_{M}\left(t\right)-E.\left|{rl.Y}_{M}\left(t\right)-Prey\left(t\right)\right|$$
(40)$$Y_{2}\left(t\right)=Y_{FM}\left(t\right)-E.\left|{rl.Y}_{FM}\left(t\right)-Prey\left(t\right)\right|$$

Finally, the position vector $Y_{1}$ is used as the optimal path.

#### 4.2.2. Fitness Function of GJO

The GJO method builds the fitness function using the following parameters in order to find the optimal route between CH and BS:(1)Distance between CH and BS: The further the distance between the CH and the BS, the more energy is required to transmit the data. Therefore, the CH close to the BS is selected to form the best path. Consequently, minimizing the objective function $f_{1}$ is our aim:
(41)$$f_{1}=1/K\sum_{\left(i=1\right)}^{K}D\left({CH}_{i},BS\right)$$(2)The ratio of the distance between CH and BS to the inter-cluster head distance:The fitness function’s second parameter will be this ratio. Here, maximizing the inter-cluster distance and minimizing the distance between ${CH}_{i}$ and BS are the main objectives. Consequently, minimizing the objective function $f_{2}$ is our aim:
(42)$$f_{2}=\left(\sum_{\left(i=1\right)}^{K}D\left({CH}_{i},BS\right)\right)/\left(\sum_{\left(i=1,j=1\right)}^{K}D\left({CH}_{i},{CH}_{j}\right)\right)$$(3)Residual energy of CH: This is the fitness function’s third parameter. Our goal is to prioritize the CH with more remaining energy as the next hop. Let $E_{res}\left({CH}_{i}\right)$ denote the residual energy of CH; *K* denotes the number of CHs. Consequently, our goal is to maximize the objective function $f_{3}$ as follows:
(43)$$f_{3}=\sum_{i=1}^{K}E_{res}\left({CH}_{i}\right)$$The following three functions $f_{1}$, $f_{2}$, and $f_{3}$, are to be minimized as the fitness function for our suggested GJO method. As a result, the following is the fitness function for our algorithm:
(44)$$f=\omega_{1}\times f_{1}+\omega_{2}\times f_{2}+\omega_{3}\times \frac{1}{f_{3}}$$
where $\omega_{1}$, $\omega_{2}$, and $\omega_{3}$ are constants that give the objective functions weights. The total of these constants ought to equal 1. Minimizing the objective function is our aim.

### 4.3. The Pseudo-Code of the Proposed Algorithm

Algorithm 1 is a pseudo-code that uses the SO algorithm to select the best set of CHs and Algorithm 2 is a pseudo-code that uses the GJO method to find the best route between CH and BS.
**Algorithm 1** SO-Clustering**Input:** Set of sensor nodes: $S=\left\{s_{1},s_{2},s_{3},\cdots ,s_{n}\right\}$**Output:**  Optimal positions of cluster heads $CH=\left\{{CH}_{1},{CH}_{2},{CH}_{3},\cdots ,{CH}_{m}\right\}$ Step1: Initialize population *X*; Step2: **for** i = 1:*N* **do**  Calculate $Fitness\left(X_{i}\right)$ using Equations (24)–(27); **end for** Take the minimum value of Fitness as Gbest; Take the individual whose Fitness value is Gbest as $X_{food}$; /*Diving the swarm into two equal groups males and females*/ Define $N_{m}=\frac{N}{2}$ as the male individual numbers; Define $N_{f}=N-N_{m}$ as the female individual numbers; **for** i = 1:*N* **do**  Calculate ${Fitness}_{m}\left(X_{i,m}\right)$; **end for** **for** i = 1:*N* **do**  Calculate ${Fitness}_{f}\left(X_{i,f}\right)$; **end for** Take the minimum value of ${Fitness}_{m}$ as $f_{best,m}$; Take the minimum value of ${Fitness}_{f}$ as $f_{best,f}$; Take the male individual whose ${Fitness}_{m}$ value is $f_{best,m}$ as $X_{best\_m}$; Take the female individual whose ${Fitness}_{f}$ value is $f_{best,f}$ as $X_{best\_f}$; Step3: **while** 
$t<t_{max}$ 
**do**  Define Temp using Equation (Equation 11);  Define food Quantity *Q* using Equation (Equation 12);  **if** 
Q<0.25 
**then**   /*Exploration phase (no food)*/   **for** i = 1:$N_{m}$ **do**    Define $A_{m}$ using Equation (Equation 7);    Update position of $X_{new\_m}$ using Equation (Equation 6);   **end for**   **for** i = $N_{f}$:*N* **do**    Define $A_{f}$ using Equation (Equation 9);    Update position of $X_{new\_f}$ using Equation (Equation 8);   **end for**  **else**   /*Exploitation phase (food exists)*/   **if** 
Temp>0.6 
**then**    **for** i = 1:$N_{m}$ **do**     Using the Brownian motion function to replace the constant coefficients $C_{3}$,     Update position of $X_{i,new\_f}$ using Equation (Equation 13);    **end for**    **for** i = $N_{F}:N$ **do**     Using the Brownian motion function to replace the constant coefficients $C_{3}$,      Update position of $X_{i,new\_f}$ using Equation (Equation 13);    **end for**   **else**    **if** 
rand>0.6 
**then**     /*Fight Mode*/     **for** i = 1:$N_{m}$ **do**      Define FM using Equation (Equation 16);      Using the Brownian motion function to replace the constant coefficients $C_{3}$,      Update position of $X_{new\_m}$ using Equation (Equation 14);     **end for**     **for** i = $N_{f}$:*N* **do**      Define FM using Equation (Equation 17);      Using the Brownian motion function to replace the constant coefficients $C_{3}$,      Update position of $X_{new\_f}$ using Equation (Equation 15);     **end for**    **else**     /*Mating Mode*/     **for** i = 1:$N_{m}$ **do**     Define $M_{m}$ using Equation (Equation 20);     Using the Brownian motion function to replace the constant coefficients $C_{3}$,      Update position of $X_{new\_m}$ using Equation (Equation 18);     **end for**     **for** i = $N_{f}$:*N* **do**      Define $M_{f}$ using Equation (Equation 21);      Using the Brownian motion function to replace the constant coefficients $C_{3}$,      Update position of $X_{new\_f}$ using Equation (Equation 19);     **end for**    **end if**    Change the worst male and female Equations (22) and (23);   **end if**  **end if**  **if** 
$f_{best,m}<\;f_{best,f}$ 
**then**   $Gbest=f_{\left(best,m\right)}$;   $X_{food}=X_{best\_m}$;  **else**   $Gbest=f_{\left(best,f\right)}$;   $X_{food}=X_{best\_f}$;  **end if**  t=t+1; **end while** Step4: Repeat Step3 until it reaches the maximum number of iterations; Step5: Return best solution $X_{food}$ /*Optimal cluster head*/
**Algorithm 2** GJO-Routing**Input:** Set of cluster heads $CH=\left\{{CH}_{1},{CH}_{2},{CH}_{3},\cdots ,{CH}_{m}\right\}$**Output:** Route *R* Initializing prey population *Y* using Equations (28)–(30); Step2: **while** 
$t<t_{max}$ 
**do**  **for** i = 1:*N* **do**   Calculate $Fitness\left(Y_{i}\right)$ using Equations (41)–(44);  **end for**  Sort the Fitness;  The first value of Fitness as $f_{1}$ (Male Jackal position);  Take the prey whose Fitness value is $f_{1}$ as $Y_{1}$;  The second value of Fitness as $f_{2}$ (FeMale Jackal position);  Take the prey whose Fitness value is $f_{2}$ as $Y_{2}$;  **for** i = 1:*N* **do**   Define the Evading Energy *E* of prey using Equations (33)–(35);   Define rl using Equations (36) and (37);   **if** E>1)  **then**    Update the position of male jackal $Y_{1}$ using Equation (Equation 31);    Update the position of female jackal $Y_{2}$ using Equation (Equation 32);    Update jackal position $Y_{new}$ using Equation (Equation 38);   **else**    Update the position of male jackal $Y_{1}$ using Equation (Equation 39);    Update the position of female jackal $Y_{2}$ using Equation (Equation 40);    Update jackal position $Y_{new}$ using Equation (Equation 38);   **end if**  **end for**  **for** i = 1:*N* **do**  **if** 
$Fitness\left(Y_{\left(i,new\right)}\right)<Fitness\left(Y_{i}\right)$ 
**then**   $Fitness\left(Y_{i}\right)=Fitness\left(Y_{\left(i,new\right)}\right)$;   $Y_{i}=Y_{\left(i,new\right)}$;  **end if** **end for** Sort the Fitness; $Y_{1}=bestprey$ t=t+1;**end while**Step 3: Repeat Step 2 until it reaches the maximum number of iterations;Step 4: Return $Y_{1}$, $\left.R\leftarrow \left\{R+Y_{i,1}\right\}\right.$

## 5. Simulation Results

### 5.1. Simulation Settings

Using the 100 × 100, 200 × 200, 300 × 300 (m^2^) setup area range for the experiment, compare the clustering impact of the technique proposed with the other three comparative algorithms. The following algorithms are compared with the suggested approaches: PSO-LSA [29], LEACH-IACA [22], and LEACH-ANT [23]. Identical sets of network input setup parameters are used by all methods. The simulation related parameters are shown in Table 1.

### 5.2. Residual Energy

This performance metric is the sum of the remaining energy of the surviving nodes in each round. For the length of the simulation, a fixed initial energy of 0.5 J has been used. The ESO-GJO technique presented in this research is compared with PSO-LSA, EPOA-CHS, LEACH-IACA, and LEACH-ANT residual energy for a scenario with 100 nodes. The results are displayed in Figure 3a. It is noted that LEACH-IACA and LEACH-ANT lost their remaining energy after 1500 cycles. On the other hand, PSO-LSA,EPOA-CHS, and ESO-GJO managed to live with 15.29 J, 16.41 J, and 18.13 J, respectively. All nodes of PSO-LSA and EPOA-CHS die after the 2400th round. However, the proposed ESO-GJO still has 0.08 J of energy remaining. Figure 3b shows the results of comparing the residual energy of ESO-GJO with other protocols for the scenario of 200 nodes. It is observed that at the end of 1000 rounds, all the energies of LEACH-IACA and LEACH-ANT are exhausted. However, PSO-LSA, EPOA-CHS, and ESO-GJO have 55.30 J, 55.01 J, and 56.79 J remaining, respectively. With PSO-LSA, all nodes die at round 2500. However, the proposed ESO-GJO still has 2.67 J of energy remaining. Figure 3c shows the results of comparing the residual energy of ESO-GJO with other protocols for the scenario of 300 nodes. It is observed that at the end of 800 rounds, all the energies of LEACH-IACA and LEACH-ANT are exhausted. However, PSO-LSA,EPOA-CHS, and ESO-GJO have 94.14 J, 94.95 J, and 95.40 J remaining, respectively. With PSO-LSA and EPOA-CHS, all nodes die at round 2350. However, the proposed ESO-GJO still has 0.05 J of energy remaining. Figure 3a–c show that the proposed ESO-GJO increases network residual energy over the compared LSA, LEACH-IACA, and LEACH-ANT protocols.

### 5.3. Network Lifetime

This section compares the network lifetimes of the PSO-LSA,EPOA-CHS, LEACH-IACA, and LEACH-ANT algorithms with the suggested ESO-GJO algorithm. The time span from the sensor network’s inception to the death of the final node is referred to in this study as the network lifetime. Four different metrics, FND (First Node Death), HND (Half Nodes Death), MND (Most Nodes Death), and LND (Last Node Death), are analyzed for different protocols. Table 2 records the number of rounds in which each indicator occurs in different scenarios for ESO-GJO, PSO-LSA, EPOA-CHS, LEACH-IACA, and LEACH-ANT. Figure 4a–c shows more visually the results of ESO-GJO compared to other protocols for the four metrics in the section above for scenarios with 100 nodes, 200 nodes, and 300 nodes. Consequently, our suggested ESO-GJO can more effectively increase the network lifetime.

### 5.4. Live Nodes Number

The total lifespan of a WSN is indicated by the number of live nodes in the network. The proposed ESO-GJO protocol and the other three protocols are compared for the number of surviving nodes every round for the scenarios of 100, 200, and 300 nodes during the course of the network lifetime in Figure 5a–c. In the case of 100 sensors, the proposed ESO-GJO algorithm extends the network lifetime by about 8.14%, 5.24%, 119.87%, and 135.76%, respectively, compared to other routing protocols. In the case of 200 sensors, the proposed ESO-GJO algorithm extends the network lifetime by about 22.3%, 8.46%, 192.2%, and 180.6%, respectively, compared to other routing protocols. In the case of 300 sensors, the proposed ESO-GJO algorithm extends the network lifetime by about 16.9%, 4.97%, 181.5%, and 363.5%, respectively, compared to the other three protocols. As a result, the ESO-GJO protocol’s suggested approach can successfully lower energy dissipation and increase the network’s lifespan.

### 5.5. Packet Delivery

The total number of packets that the BS has received is the definition of this network performance indicator. Regarding network throughput, the ESO-GJO approach performs better than the LSA, LEACH-IACA, and LEACH-ANT algorithms in Figure 6a (100 nodes). Specifically, ESO-GJO’s network throughput outperformed PSO-LSA,EPOA-CHS, LEACH-IACA, and LEACH-ANT by 7.89%, 5.44%, 119.3%, and 133.1%, respectively. The ESO-GJO algorithm outperforms PSO-LSA,EPOA-CHS, LEACH-IACA, and LEACH-ANT in terms of network throughput when Figure 6b (200 nodes) is considered. Specifically, ESO-GJO’s network throughput outperformed PSO-LSA,EPOA-CHS, LEACH-IACA, and LEACH-ANT by 6.62%, 4.70%, 167.36%, and 156.22%, respectively. Finally, the ESO-GJO algorithm outperforms PSO-LSA, EPOA-CHS, LEACH-IACA, and LEACH-ANT in terms of network throughput when Figure 6c (300 nodes) is considered. Specifically, ESO-GJO’s network throughput outperformed PSO-LSA, EPOA-CHS, LEACH-IACA, and LEACH-ANT by 1.02%, 0.58%, 308.26%, and 149.70%, respectively. These outcomes unequivocally demonstrate the improved network throughput that the ESO-GJO algorithm achieves in many contexts as well as its superiority over the other protocols.
Figure 4Network life cycle. Scenarios with (**a**) 100 nodes; (**b**) 200 nodes; (**c**) 300 nodes.
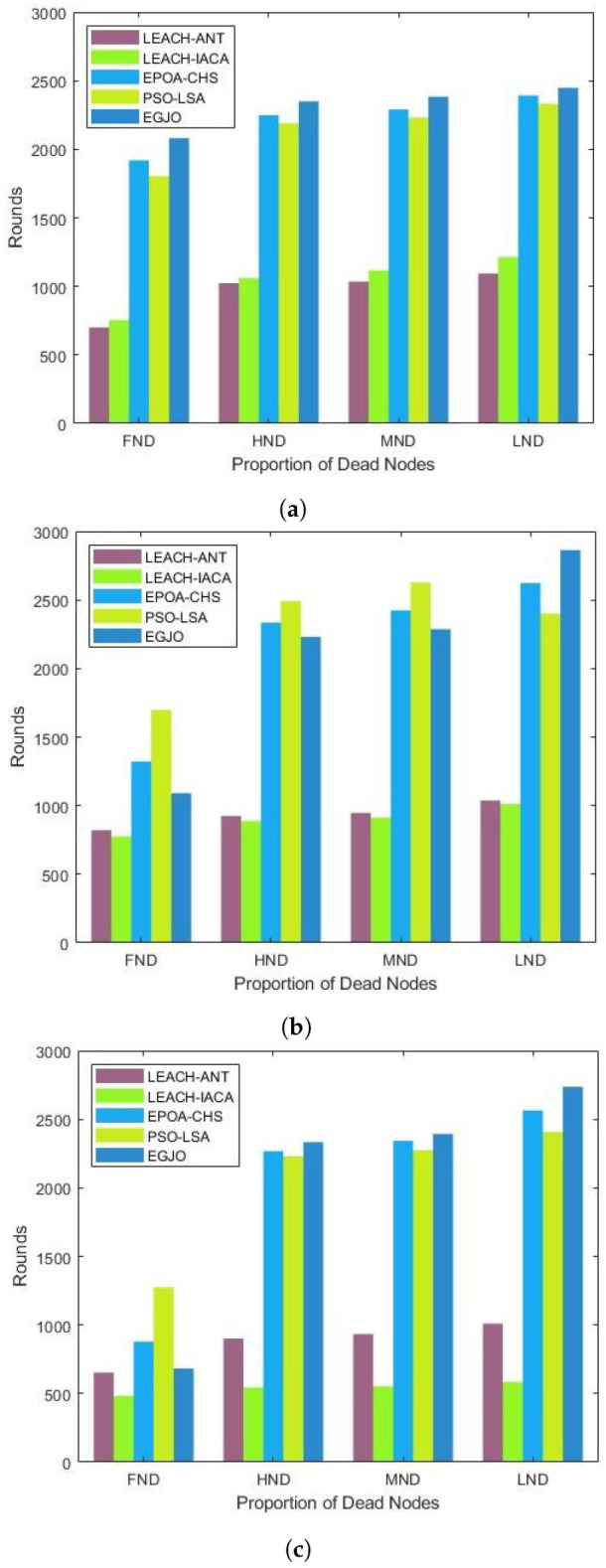

Figure 5Live nodes number in each round. Scenarios with (**a**) 100 nodes; (**b**) 200 nodes; (**c**) 300 nodes.
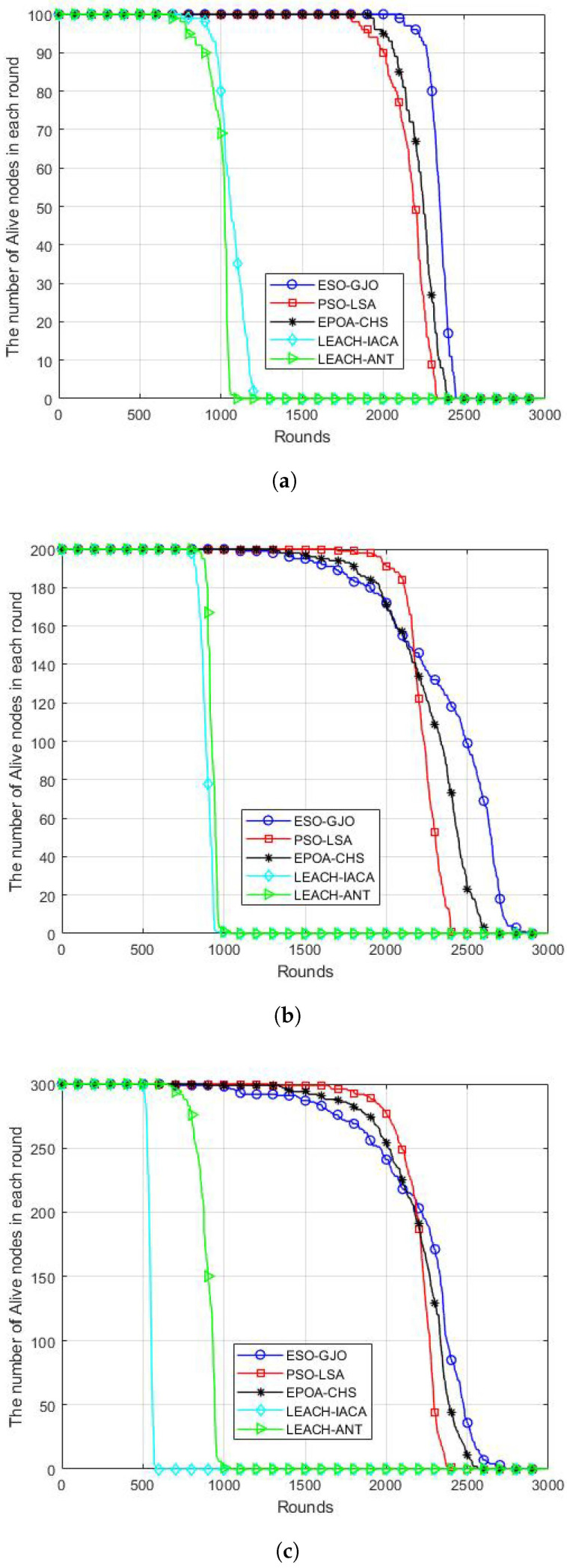

Figure 6Packet delivery in each round. Scenarios with (**a**) 100 nodes; (**b**) 200 nodes; (**c**) 300 nodes.
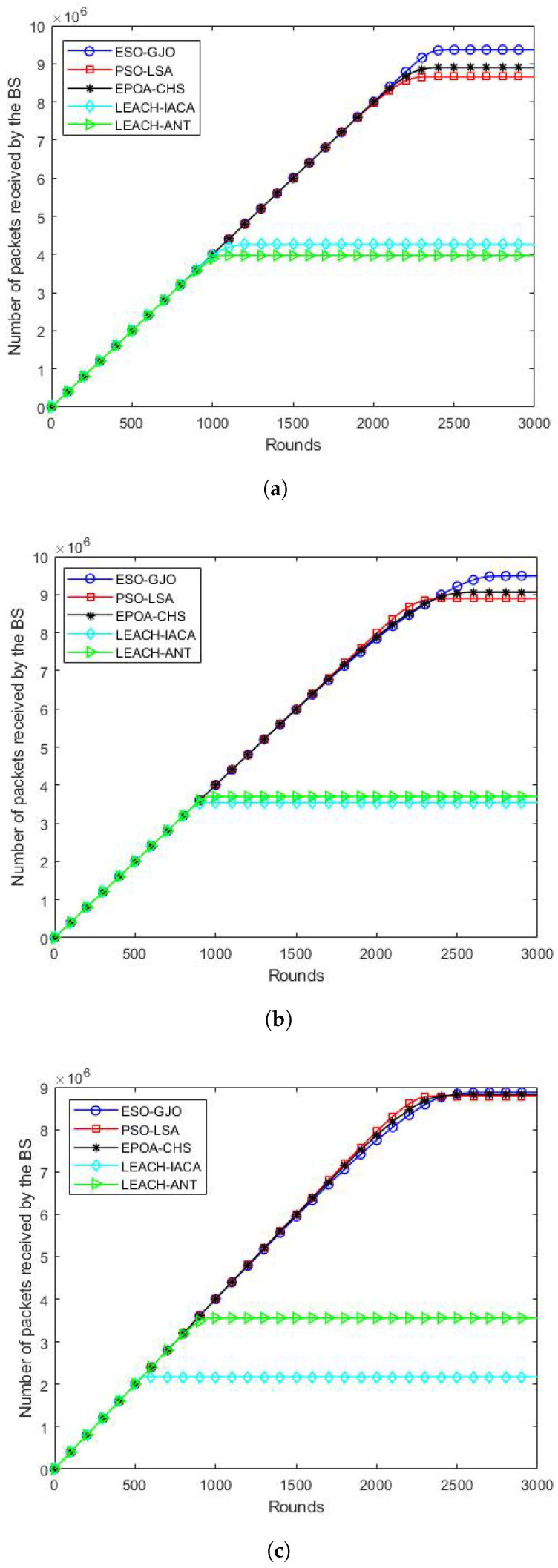



### 5.6. Energy Consumption

This network performance metric displays the total energy utilized by all of the active nodes in the network throughout each cycle. Based on simulation results, Figure 7a–c illustrate that the suggested ESO-GJO model outperforms PSO-LSA,EPOA-CHS, LEACH-IACA, and LEACH-ANT. By taking the objective function’s energy impact into account, the ESO-GJO algorithm reduces network energy consumption and lengthens the network’s lifespan.

## 6. Conclusions

We have created a novel ESO-GJO method for energy-constrained WSNs in this study. In order to minimize energy usage, we construct an objective function and select the set of optimal CHs. The experimental results have been published together with a comparison of the LSA, LEACH-IACA, and LEACH-ANT algorithms that are currently in use. In the test scenarios with 100, 200, and 300 sensor nodes, respectively, the algorithm is run. The results indicate that, in comparison to alternative protocols, the strategy outlined in this study is successful in reducing network energy consumption and improving network lifetime. However, the algorithm still has limitations; the performance metric FND falls short in large-scale networks. With the aforementioned restrictions in mind, future research can build on these findings to prolong the FND emergence period for large-scale WSNs.

## Figures and Tables

**Figure 1 sensors-24-01348-f001:**
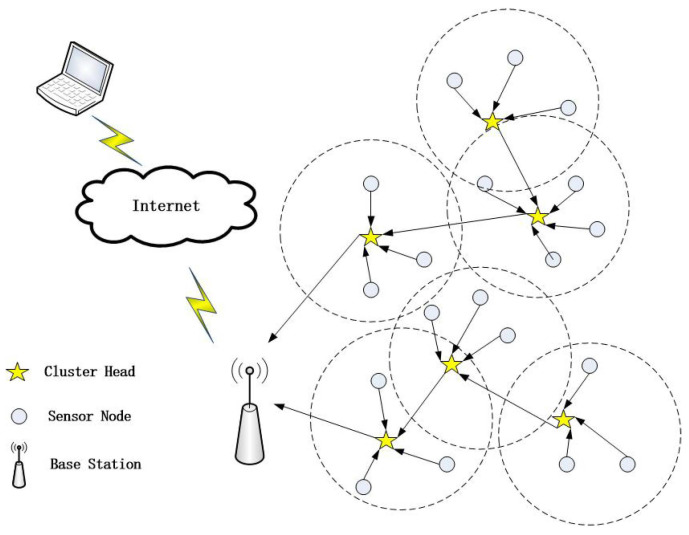
Cluster-based WSN with CHs.

**Figure 2 sensors-24-01348-f002:**
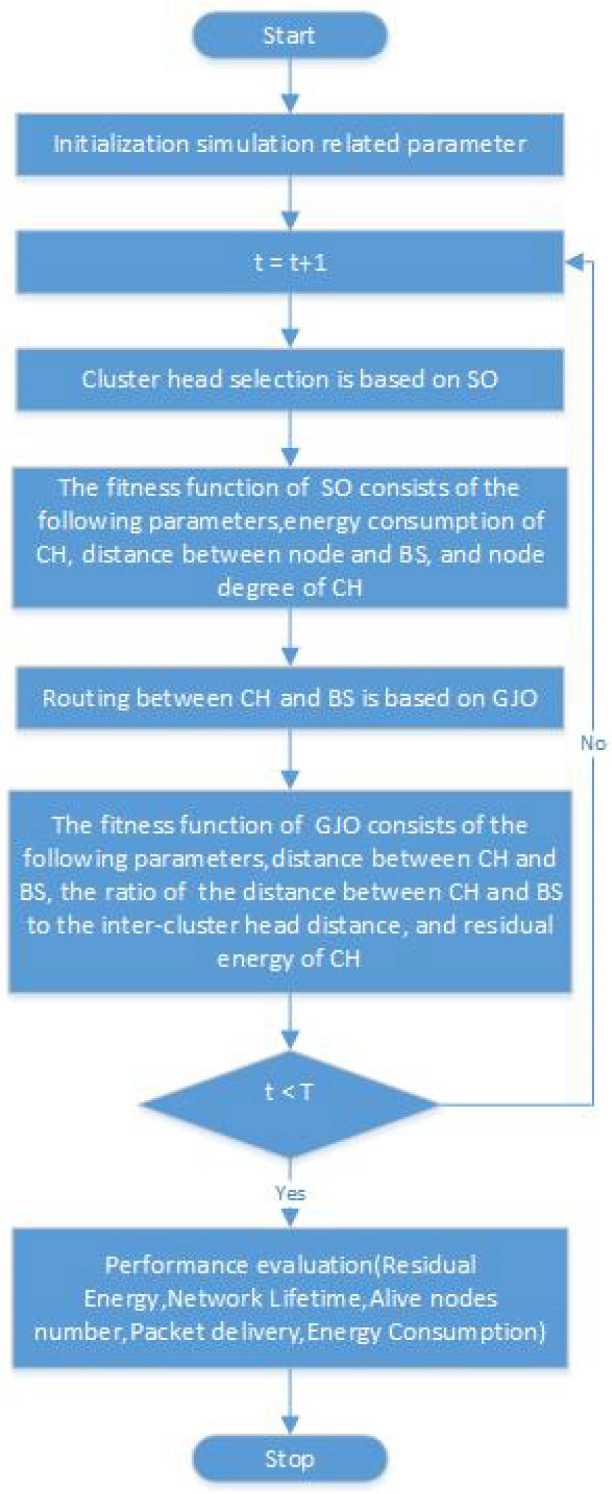
ESO-GJO flowchart.

**Figure 3 sensors-24-01348-f003:**
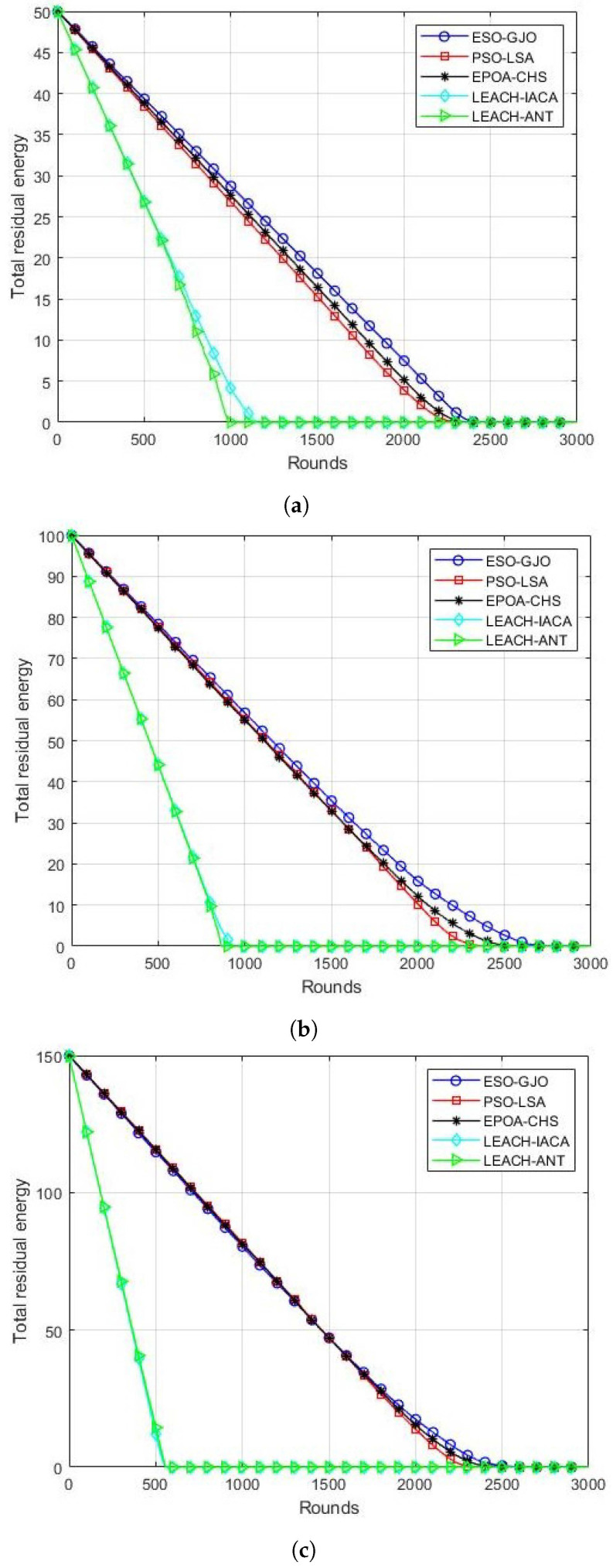
Residual energy in each round. Scenarios with (**a**) 100 nodes; (**b**) 200 nodes; (**c**) 300 nodes.

**Figure 7 sensors-24-01348-f007:**
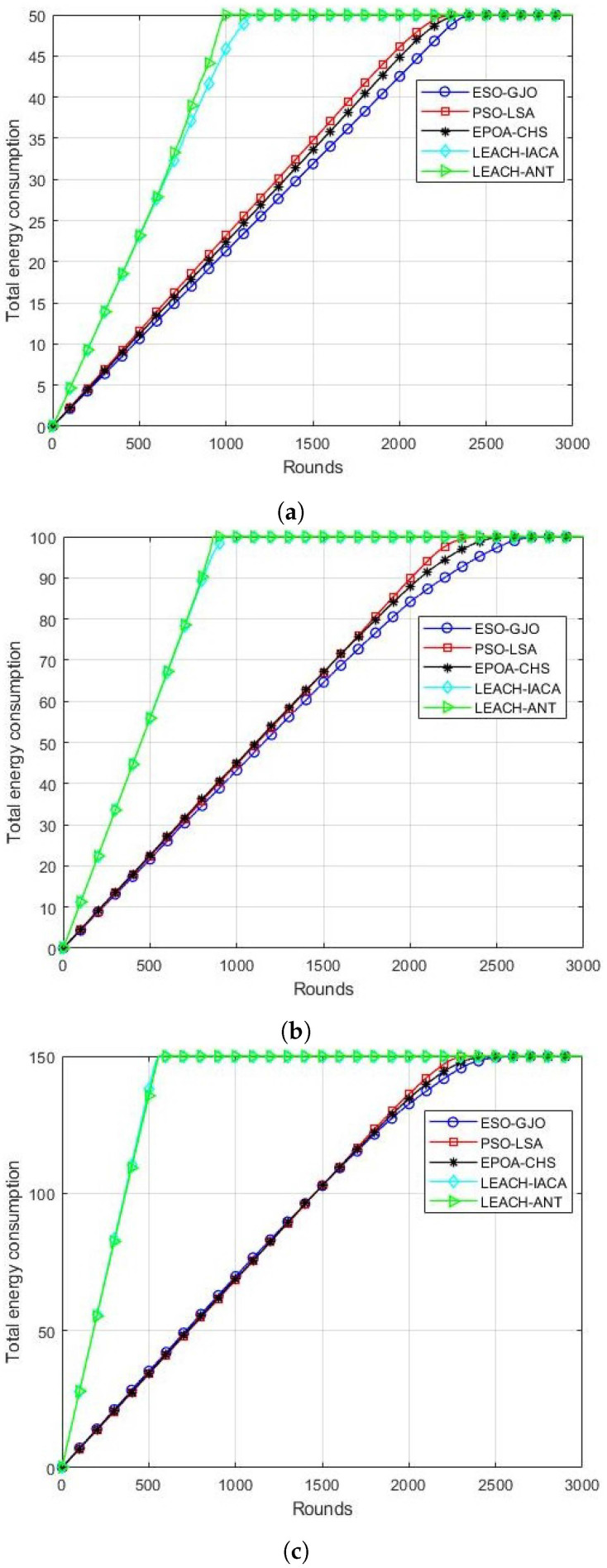
Energy consumption in each round. Scenarios with (**a**) 100 nodes; (**b**) 200 nodes; (**c**) 300 nodes.

**Table 1 sensors-24-01348-t001:** Simulation-related parameter.

Parameters	Value
Network Area	100 × 100, 200 × 200, 300 × 300 (m^2^)
Number of nodes	*N* = 100, 200, 300
Sink node location	(50 m, 150 m)
	(100 m, 250 m)
	(150 m, 350 m)
$E_{0}$	0.5 J
Packet Size	4000 Bits
$E_{elec}$	50 nJ/bit
$\epsilon_{fs}$	10 nJ/bit/m^2^
$\epsilon_{mp}$	0.0013 pJ/bit/m^4^
$E_{DA}$	5 nJ/bit/signal
$d_{0}$	70 m

**Table 2 sensors-24-01348-t002:** Round and lifetime.

No. of Rounds					
**Simulation Scenario**	**Protocol**	**FND**	**HND**	**MND**	**LND**
Number of nodes is 100	ESO-GJO	2080	2350	2383	2448
	PSO-LSA	1805	2191	2233	2332
	EPOA-CHS	1920	2249	2290	2392
	LEACH-IACA	755	1060	1116	1214
	LEACH-ANT	701	1023	1035	1094
Number of nodes is 200	ESO-GJO	1091	2231	2287	2863
	PSO-LSA	1698	2492	2628	2402
	EPOA-CHS	1322	2334	2424	2623
	LEACH-IACA	775	889	913	1012
	LEACH-ANT	821	925	947	1038
Number of nodes is 300	ESO-GJO	683	2333	2393	2737
	PSO-LSA	1275	2230	2275	2407
	EPOA-CHS	879	2268	2343	2565
	LEACH-IACA	483	546	553	584
	LEACH-ANT	653	901	934	1010

## Data Availability

Data is contained within the article.

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
