# Peer review of "Multi-Hop Clustering and Routing Protocol Based on Enhanced Snake Optimizer and Golden Jackal Optimization in WSNs"

_sensors, 2024, doi:10.3390/s24041348_

Round 1
Reviewer 1 Report
Comments and Suggestions for Authors
- There exist a wide range of optimization strategies and the selection of Snake Optimizer and Golden Jackal Optimization is not justified. The benefits of using these strategies in the problem in hand should be supported by an extensive analysis of the state of the art in optimization. Otherwise, its selection seems to be arbitrary.
- The paper presents comparative results to evaluate the validity of the proposal. Nevertheless, no all the selected strategies for comparison (e.g., LSA) are supported by the previous analysis done in the related work. Why have the considered strategies been selected for comparison after the analysis of previous works? Are these strategies recent?
- The simulation setup is not clear. More details about the considered simulator are necessary. The results are not reproducible in the current form.
- Also, why the are the presented values for parameters have been chosen? How can you justify them? How can you guarantee a valid range of condition to support the quality of the proposal?
- Figures in experimental section such as Figure 4, 5, 6, 7 and 8 are too small and the results cannot be appropriately analyzed.
Reviewer 2 Report
Comments and Suggestions for Authors
"Given that the primary cause of node energy consumption is wireless communication" - please provide a reference to a source that proves this statement.
The algorithms require some general description to explain the idea first and details later. Thus they will become more clear. Currently the paper lacks some connection between the entities of the wireless network and the optimizator entities.
Figures 7-9 are not informative and are hard to understand and compare. I suggest replacing them with something more easily tractable. Are they really needed to support some conclusions made in the paper?
Comments on the Quality of English LanguageThe paper requires proodreading, because it contains grammatical errors. For example: "WSNs is a network made up of ..." should be either "WSN is..." or "WSNs are networks...".
Author Response
Response to Reviewer Comments
Point 1: There exist a wide range of optimization strategies and the selection of Snake Optimizer and Golden Jackal Optimization is not justified. The benefits of using these strategies in the problem in hand should be supported by an extensive analysis of the state of the art in optimization. Otherwise, its selection seems to be arbitrary.
Point 1: "Given that the primary cause of node energy consumption is wireless communication" - please provide a reference to a source that proves this statement.
Response 1:
Thank you for your valuable advice. We have followed your suggestion to add a citation to the source for this statement, which is identified in yellow in the paper.
Point 2: The algorithms require some general description to explain the idea first and details later. Thus they will become more clear. Currently the paper lacks some connection between the entities of the wireless network and the optimizator entities.
Response 2:
I would like to thank the reviewers and editors for their valuable opinions on this paper. We have followed your suggestion and added a generalized description of the algorithm, identified in yellow in the paper
Point 3: Figures 7-9 are not informative and are hard to understand and compare. I suggest replacing them with something more easily tractable. Are they really needed to support some conclusions made in the paper?
Response 3:
Thank you for your valuable advice. We redid the simulation experiments as you suggested, and here are the diagrams that replace Figures 7-9.
Point 4: The paper requires proodreading, because it contains grammatical errors. For example: "WSNs is a network made up of ..." should be either "WSN is..." or "WSNs are networks...".
Response 4:
Thank you for your valuable advice. We have made the appropriate changes in the paper as per your suggestions and have marked them in yellow color

Round 2
Reviewer 2 Report
Comments and Suggestions for Authors
The authors have resolved my comments and improved the paper. I think it is fine.
